

# The impact of combined application of biochar and fertilizer on the biochemical properties of soil in soybean fields

Mingcong Zhang[1,2], Wei Xie[1], Xingjie Zhong[1], Yuqing Wang[1], Siyan Li[1], Yanhong Zhou[1] and Chen Wang[1]

[1] College of Agronomy, Heilongjiang Bayi Agricultural University, Daqing, China
[2] Key Laboratory of Low-carbon Green Agriculture in Northeastern China, Ministry of Agriculture and Rural Affairs P. R. China

## ABSTRACT

**Background:** Heilongjiang Province is a major soybean production area in China. To improve soil structure and increase soybean yield, this study examined the effects of combined biochar and chemical fertilizer application on the biochemical properties of soil in a maize-soybean rotation system.

**Methods:** The research were conducted from 2021 to 2022 at Heshan Farm Science Park in Heilongjiang Province, this field plot experiment utilized two soybean varieties, Heihe 43 (a high-protein variety) and Keshan 1 (a high-oil variety). In 2021, two plots with similar fertility levels were selected for planting soybeans and maize. In 2022, a maize-soybean rotation was implemented with five treatments: conventional fertilization (CK), increased biochar+reduced fertilizer 1 (F1+B), reduced fertilizer 1 (F1), increased biochar+reduced fertilizer 2 (F2+B), and reduced fertilizer 2 (F2). The study systematically analyzed the effects of combined biochar and chemical fertilizer application on soil chemical properties and microbial characteristics.

**Results:** Over 2 years, results showed that combined application effectively improved soil chemical traits. Compared to conventional fertilization (CK) and reduced fertilization (F1, F2), t he combined application of biochar and chemical fertilizer (F1 +B, F2+B) increased soil pH, EC and the absolute value of zeta potential of soil surface, the CEC of soil significantly increased by 15.6–44.3%, the soil surface charge density and the soil surface charge quantity significantly increased by 16.4–73.5%. The combined application of biochar and chemical fertilizer also effectively enhanced the abundance and diversity of soil microbes. Dominant bacterial groups in soybean field soils under different treatments included Actinobacteria, Acidobacteria, Chloroflexi, and Proteobacteria; while dominant fungal groups were Ascomycota, Basidiomycota, and Mortierellomycota. Alpha and Beta diversity analyses revealed that the F1+B treatment significantly enhanced the species richness and diversity of bacteria and fungi in the soil, increasing the proportion and evenness of dominant and beneficial genera.

Corresponding authors
Yanhong Zhou,
2110447789@qq.com
Chen Wang, 15147055121@163.com

## INTRODUCTION

Soil chemical properties change with field management practices and crop growth processes. Traditionally, fertilization is thought to cause excessive accumulation of nitrate nitrogen in soil, leading to soil acidification, accelerated leaching of nutrients, and depletion of essential elements, which affects crop growth. Rapid and significant accumulation of soil nitrates can lead to salinization and secondary salinization, causing soil hardening and disrupting stable aggregate structures, thereby reducing crop yields and quality (*Li, Tan & He, 2009*; *Li, Hu & Cheng, 2003*; *Zhang, 2018*). Biochar, an organic material rich in aromatic structures, regulates farmland soil effectively (*Vida, Fayez & Mohammad, 2016*). Biochar can improve soil properties, increases soil fertility, and enhances plant stress resistance, thereby boosting crop yields and agricultural productivity. Studies by *Mendes et al. (2015)* and *Wang et al. (2018)* have shown that the application of rice husk biochar promoted the increase of 0.25 to 5 mm large aggregates, decreased the content of <0.25 mm micro-aggregates, and significantly increased the proportion of MWD and GMD in soil. This provides a larger adsorption area for the adhesion of nutrient elements, enhances the electrostatic adsorption force between soil particles, and improves the water stability of soil aggregates. Due to its high porosity, biochar also increases soil porosity, reducing soil bulk density and enhancing water infiltration and water-holding capacity. Previous studies have shown that the three ratios of soil applied with biochar are close to the optimal value, which enhances soil water infiltration and water holding function, improves soil pH which is reduced due to long-term single application of chemical fertilizer in successive years, improves the cation exchange capacity of soil solution, and provides a more suitable soil microecological environment for crops and soil microbial activities. This will promote the decomposition of plant and animal residues, improve the mineralization and accumulation of nitrogen in the soil, promote the synthesis of organic matter in the soil, so that the soil can retain more nutrients, increase the soil's fertilizer retention capacity, and also improve the water stability of the soil, thereby improving the soil nutrient retention (*Yang, 2022*; *Yuan, 2018*). The combined application of biochar and chemical fertilizer promotes nutrient retention in soil, accelerates soil nutrient transformation, replenishes nutrients lost due to reduced fertilization, and increases the total nitrogen content and inorganic nitrogen content in the tillage layer (*Li, 2015*). Soil colloids are the main site for various chemical and biochemical processes in soil and contain most of the negative charge in soil components, making them the finest and most active constituents in soil (*Erika et al., 2023*; *Wei et al., 2023*). Soil electrochemical characteristics significantly influence soil properties and functions, determining the rate of material transformation, and nutrient and water uptake in soil, thus providing important guidance for agricultural production. Studies have found that the electrochemical properties of soil colloids hinder the migration of heavy metals in fertilizers, mainly due to: spatial hindrance, blocking effects caused by the large particle size of mobile colloid particles adsorbing heavy metals, and heavy metals being adsorbed or complexed by immobile colloids (*Yang, 2022*; *Lun et al., 2023*). Research by *Wang (2010)* on saturated sandy loam soil columns has shown that the presence of $SiO_2$ colloids

increases the adsorption of Cd in soil compared to soil without colloids, increases the number of adsorption sites on the soil surface, and inhibits the migration of Cd in the soil column. Research by *Yang & Lu (2020)* has shown that straw biochar application improves soil surface electrochemical characteristics, enhances soil surface potential, and stably increases soil cation exchange capacity. However, on the other hand, the production of biochar requires high energy consumption, and large-scale production of biochar needs a large amount of biomass, which can leave more heavy metals and harmful substances, potentially polluting the environment and causing a decline in ecosystem diversity. Thus, while biochar has great potential for improving soil physical and chemical characteristics, current research in China mainly focuses on combining chemical and organic fertilizers to reduce fertilizer use and enhance effectiveness. How to utilize the advantages of biochar in different regions and under different crop rotation backgrounds to improve the resistance and stability of soil ecosystems still warrants further investigation.

A study in an agricultural ecosystem involving the application of inorganic fertilizers to clay loam soils (saturated immature soils) for 50 years showed significant reductions in soil pH, soil acidification, and notable decreases in microbial diversity and activity (*Pereira et al., 2018*), severely damaging soil microbial abundance and diversity. However, biochar, with its rich alkaline functional groups, can increase soil pH. Its high specific surface area and rich carbon source can provide ample moisture and nutrients for soil microbes, promoting the enhancement of microbial abundance and diversity and protecting the stability of the ecosystem (*Lehmann et al., 2011*). Other studies have shown that the richness of microbes is closely linked to the structure of soil aggregates. Research indicates that soil aggregates of different sizes, due to variations in porosity, oxygen, and nutrient conditions, exhibit differing microbial distributions; larger aggregates tend to have more fungal communities, while smaller aggregates are more populated by bacterial communities (*Jagan & Neelakantan, 2023*). A good aggregate structure not only provides suitable living conditions for microbes but also the secretions from the microbes themselves can bind soil colloids to form robust soil aggregate structures (*Lei et al., 2017*). Therefore, exploring the impact of biochar on soil microbial characteristics often requires comprehensive and systematic research. Most studies suggest that compared to traditional fertilization, the co-application of biochar and chemical fertilizers can effectively increase soil microbial abundance and diversity, and optimize the soil microbial community structure (*Ye et al., 2023*); some research has found that reduced nitrogen combined with biochar promotes the growth of bacteria and fungi in dry red soil (*Xie et al., 2023*). An active microbial community can accelerate its own organic system metabolism, increase soil enzyme activity, hasten the decomposition of plant and animal residues and microbial remains, and promote the synthesis of humus, improving soil nutrients. In the measures combining biochar and chemical fertilizers, by providing more carbon and nitrogen sources to the soil and increasing soil porosity, there is a significant increase in the abundance and diversity of microbial communities (*Rijk et al., 2024*). Additionally, bio-oil and unstable compounds adsorbed on the surface of biochar during its production can serve as substrates for microbial metabolic reactions. Short-term structural changes in the soil after biochar application, such as surface oxidation, can provide some carbon sources

for microbial growth and reproduction (*Wang et al., 2022*), showing that biochar not only improves the physical and chemical characteristics of the soil but also indirectly provides suitable living conditions for soil microbes. Moreover, other studies have shown that applying biochar to the soil can increase microbial activity in the short term, with long-term application having no effect on microbial activity and even inhibiting it in some cases (*Kishimoto & Sugiura, 1985*). Simultaneously, applying biochar affects the physical (bulk density, moisture content, porosity, *etc.*) and chemical properties (pH value, cation exchange, *etc.*) of the soil, significantly impacting the composition and diversity of fungal, bacterial, and archaeal populations and leading to a reduction in the soil microbial carbon metabolism rate (*De Tender et al., 2016*). The differences in these research results are primarily dependent on different regional climates, microbial ecological structures, and soil textures. To date, there are few studies on the changes in microbial community structures under the application of biochar and chemical fertilizers both domestically and internationally.

In summary, the co-application of biochar and chemical fertilizers is significant for improving soil chemical properties, enhancing microbial diversity, and improving microbial community structures. However, the impact of biochar and fertilizer co-application on soil quality and crop growth under crop rotation conditions is still limited by local climate, environment, and soil types and requires further exploration. In recent years, due to the continuous decline of soybean planting income and the continuous decline of farmers' enthusiasm for soybean planting, the quality and yield of soybeans have been seriously affected. Therefore, it is imperative to improve the quality and yield of soybean by improving the soil properties of soybean field. This study, through field trial analysis, explores the impact of biochar and chemical fertilizer co-application on soybean field soil chemical properties and microbial diversity, seeking the most suitable method for the co-application of biochar and chemical fertilizers in Heilongjiang's soybean production areas, providing a theoretical basis for increasing crop yields and improving soil quality in the main soybean-producing areas of Heilongjiang.

# MATERIALS AND METHODS

## Experimental site overview

The experimental site is located in the Heshan Farm Science Park in Heilongjiang Province, situated between 48°43′–49°03′N latitude and 124°56′–126°21′E longitude. The region experiences a cold temperate continental climate, with an average annual temperature of 1.8 °C. The annual effective accumulated temperature ranges between 2,300 °C and 2,450 °C, the average annual sunshine duration is 2,429.3 h, and the average annual precipitation is 601.1 mm. The average frost-free period is 119 days per year. The average temperature and precipitation from April to October in 2021 and 2022 are shown in Fig. 1. The soil type of the experimental fields is chernozem, and the basic physical and chemical properties of the soil in the 0–20 cm plow layer prior to the experiment are presented in Table 1.

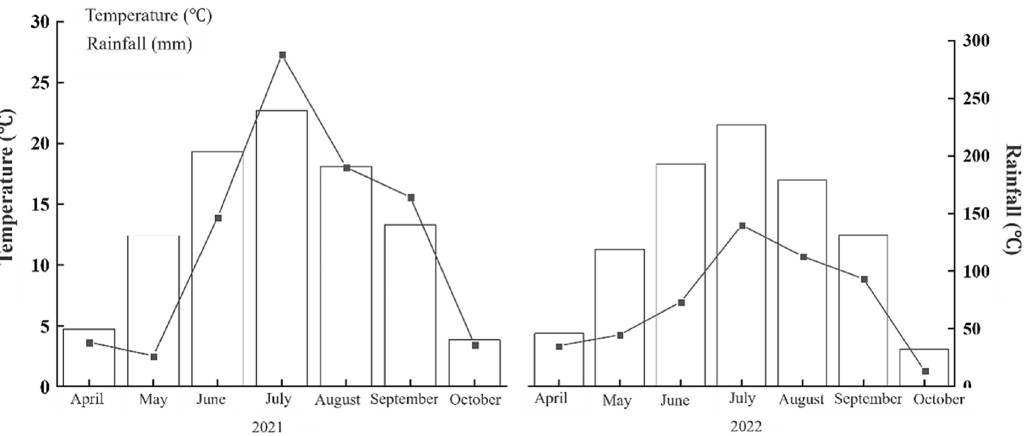

**Figure 1 Temperature and rainfall in 2021 and 2022.** The annual average temperature is 1.8 °C, the annual effective accumulated temperature is 2,300 °C–2,450 °C, and the annual average rainfall is 601.1 mm.                                   

**Table 1 Basic chemical properties of 0–20 cm surface soil before the experiment.**

| Years | pH | $NH_4$–N (mg/kg) | AP (mg/kg) | AK (mg/kg) | SOM (g/kg) | Bulk density (g/cm³) |
|---|---|---|---|---|---|---|
| 2021 | 6.20 | 137.92 | 22.54 | 174.6 | 28.51 | 1.44 |
| 2022 | 6.26 | 141.80 | 30.60 | 182.5 | 31.94 | 1.46 |

## Experimental materials

Test soybean varieties: Keshan 1 (high-oil type), exhibits semi-determinate pod-setting habit, plant height approximately 71.5 cm, main stem with 12.3 nodes, 0.2 effective branches per plant, purple flowers, and long leaves. The seeds are round with a yellow seed coat and yellow hilum, each seed weighing about 19.8 g. The crude protein content is 38.04%, and the crude fat content is 21.82% (*Zhang et al., 2013*). Heihe 43 (high-protein type), also shows a semi-determinate pod-setting habit, plant height around 75 cm, non-branching, with purple flowers, pointed leaves, and grey pubescence. The seeds are round, yellow-coated, with a light yellow hilum, glossy, each seed weighing approximately 20 g, containing 41.84% crude protein and 18.98% crude fat (*Han et al., 2016*).

Test fertilizers: Urea (containing N ≥ 46%), diammonium phosphate (containing N ≥ 18%, $P_2O_5$ ≥ 46%), and potassium sulfate (containing $K_2O$ ≥ 50%) were all purchased from Nenjiang City Fuli Agricultural Production Materials Co., Ltd. The basic physical and chemical properties of the biochar used are shown in Table 2.

## Experimental design

The experiment was conducted in the autumns of 2021 and 2022, with plowing and ridging post-harvest, followed by soil compaction and moisture retention in early spring. In 2021, the plots with soybean as the previous crop will be selected, and the corn soybean rotation planting mode will be adopted, and corn will be planted in the soybean field in 2022. The biochar application rates for the soybean fields were set at 1 and 2 t/ha (as

**Table 2 Sources and physicochemical properties of biochar.**

| Materials | Place of origin | Ingredients | Specific surface area (m²/g) | pH | Ash content (%) | C (%) | N (%) | P (%) | K (%) | CEC (cmol/kg) |
|---|---|---|---|---|---|---|---|---|---|---|
| Biochar | Mishan City in Heilongjiang Province | Corn stalk | 7.29 | 9.46 | 27.9 | 55.31 | 1.35 | 0.557 | 1.422 | 24.6 |

**Table 3 Fertilization treatment of soybean.**

| Treatments | N (kg/hm²) | P₂O₅ (kg/hm²) | K₂O (kg/hm²) | Biochar (t/hm²) |
|---|---|---|---|---|
| CK | 54 | 67.5 | 37.5 | — |
| F1+B | 40.5 | 61.95 | 23.25 | 1 |
| F1 | 40.5 | 61.95 | 23.25 | — |
| F2+B | 27 | 56.4 | 9 | 2 |
| F2 | 27 | 56.4 | 9 | — |

determined from preliminary trials), and applied into each furrow before planting, with continuous application over the 2 years.

Five treatments were established: (1) CK: conventional fertilization; (2) F1+B: reduced fertilization plus addition of biochar 1; (3) F1: reduced fertilization 1; (4) F2+B: reduced fertilization plus addition of biochar 2; (5) F2: reduced fertilization 2. Each treatment adhered to the principle of equivalent nutrient substitution, with biochar treatments reducing the amounts of nitrogen, phosphorus, and potassium normally applied in conventional fertilization. Each treatment row was 7 m long with a row spacing of 0.65 m, comprising eight rows per plot, replicated three times. The amount of conventional fertilizer applied matched local fertilization levels, and other field management practices were consistent with local production standards. Detailed fertilization amounts are shown in Tables 3.

## Measurement items and methods

Soil sampling method: Soil samples were collected from the 0–20 cm plow layer at the ridges of each plot during the soybean stages V2 (seedling), R2 (flowering), R4 (podding), R6 (grain filling), and R8 (maturity). The collected soil was gently broken into small chunks, thoroughly mixed using the quartering method, and cleared of plant debris and roots manually. Part of the fresh soil samples was air-dried indoors and sieved, another portion was stored in sealed bags at −20 °C for future use, and another part was preserved at −80 °C for microbial characteristics analysis.

Soil pH and electrical conductivity were measured using a pH meter and an electrical conductivity meter, respectively. Soil cation exchange capacity was determined using the cobaltihexamine trichloride extraction-spectrophotometric method (HJ 889-2017, 2017). Soil surface zeta potential was measured with a high-sensitivity zeta potential analyzer (zeta PALS Brookhaven, USA). The measurement of soil surface chemical properties was

performed using a combined method for material surface properties established by *Li et al. (2016)*.

For high-throughput sequencing of soil microbial characteristics, soil samples from the R8 stage of the CK, F1+B, and F1 treatments in 2022 were sent to Biomarker Technologies in Beijing. Using the Illumina Novaseq sequencing platform, a paired-end sequencing approach was used to construct a small fragment library. In this study, the highly variable V3-V4 region of bacterial 16SrRNA gene with a length of about 468 bp was selected for sequencing. The target fragment of V3-V4 region was amplified by PCR using priors 338F and 806R, and the sequence was: 338F: 5′-ACTCCTACGGGAGGCAGCAGCAG-3′/ 806R: 5′-GGACTACHVGGGTWTCTAAT-3′. In this experiment, fungal ITS1 region was selected for sequencing. PCR amplification of the target fragment in region V1 was performed using primers ITS5F and ITS1R, and the sequence was ITS5F: 5′-GGAAGTAAAAGTCGTAACAAGGG-3′/ITS1R: 5′-GCTGCGTTCTTCATCGATGC-3′ (*Fiona et al., 2022*). The samples were extracted using OMEGA Soil DNA Kit (M5366-02) (Omega Bio-Tek, Norcross, GA, USA) kit. After extracting the total DNA from the samples, primers were designed based on conserved regions, with sequencing adapters added to the ends. The samples underwent PCR amplification, and the products were purified, quantified, and normalized to form the sequencing library. After quality checks, successful libraries were sequenced using the Illumina Novaseq 6000. Based on the Pacbio third-generation sequencing platform, perform sequencing and bioinformatics analysis on the data. Use USEARCH (version 10.0) to perform OTU clustering on sequences at a similarity level of 97%, and use QIIME2 (versoin 2020.6) for 16SrRNA and ITS annotation in GenBank, abundance analysis, and alpha diversity calculation to obtain information on microbial richness and evenness within the sample. Taxonomic analysis was conducted based on the sequence composition of the Features, determining the community structure and microbial clustering at various taxonomic levels such as phylum, class, order, family, and genus. Alpha diversity analysis was performed to study microbial diversity within individual samples, calculating indices such as Ace, Chao1, Shannon, and Simpson, and drawing rarefaction and rank-abundance curves. The differences of bacterial community structure in different treatment groups were analyzed by PCA.

### Data processing

Data were processed and analyzed using Excel 2020. Variance analysis and significance testing ($P < 0.05$) were performed using SPSS 22.0 software. Multiple comparisons were made using Duncan's method, and graphs were plotted using Origin 2021.

## RESULTS

### Impact of combined biochar and chemical fertilizer application on the chemical properties of soil in soybean fields

As shown in Table 4, in 2021, the pH levels were lowest during the conventional fertilization treatment (CK), indicating that the application of chemical fertilizers reduced soil pH. The addition of biochar increased soil pH, with the F2+B treatment showing higher soil pH values than the F1+B treatment. Both biochar and chemical fertilizer

**Table 4 Effects of biochar and fertilizer on soil pH and EC in soybean planting area in 2021.**

| Periods | Varieties | Treatments | Soil pH | EC (ms/m) |
|---|---|---|---|---|
| R2 | Heihe 43 | CK | 6.13 ± 0.21b | 66.25 ± 1.11a |
| | | F1+B | 6.36 ± 0.32ab | 65.20 ± 2.34ab |
| | | F1 | 6.27 ± 0.21ab | 54.88 ± 0.77b |
| | | F2+B | 6.61 ± 0.20a | 65.45 ± 4.05ab |
| | | F2 | 6.18 ± 0.17ab | 61.81 ± 2.78b |
| | Keshan 1 | CK | 6.32 ± 0.17a | 71.25 ± 2.65a |
| | | F1+B | 6.39 ± 0.11a | 68.15 ± 1.65ab |
| | | F1 | 6.21 ± 0.16a | 64.15 ± 2.99b |
| | | F2+B | 6.37 ± 0.13a | 73.00 ± 2.84a |
| | | F2 | 6.26 ± 0.11a | 66.60 ± 1.77ab |
| R4 | Heihe 43 | CK | 6.23 ± 0.21a | 68.35 ± 1.21a |
| | | F1+B | 6.49 ± 0.23a | 66.24 ± 2.44ab |
| | | F1 | 6.33 ± 0.25a | 56.78 ± 0.87b |
| | | F2+B | 6.54 ± 0.22a | 66.75 ± 2.05ab |
| | | F2 | 6.25 ± 0.17a | 63.88 ± 2.86b |
| | Keshan 1 | CK | 6.22 ± 0.11a | 67.43 ± 1.65ab |
| | | F1+B | 6.29 ± 0.12a | 65.25 ± 3.65a |
| | | F1 | 6.11 ± 0.16a | 64.44 ± 2.76b |
| | | F2+B | 6.37 ± 0.14a | 68.20 ± 2.84a |
| | | F2 | 6.16 ± 0.12a | 63.60 ± 1.77ab |
| R6 | Heihe 43 | CK | 6.25 ± 0.12a | 98.55 ± 11.22a |
| | | F1+B | 6.37 ± 0.14a | 89.83 ± 6.67a |
| | | F1 | 6.27 ± 0.12a | 65.85 ± 4.75b |
| | | F2+B | 6.43 ± 0.16a | 91.21 ± 5.6a |
| | | F2 | 6.34 ± 0.15a | 66.34 ± 7.11b |
| | Keshan 1 | CK | 6.23 ± 0.11a | 85.33 ± 2.75a |
| | | F1+B | 6.26 ± 0.12a | 68.17 ± 4.32bc |
| | | F1 | 6.11 ± 0.1a | 69.76 ± 4.15bc |
| | | F2+B | 6.29 ± 0.13a | 74.34 ± 2.65b |
| | | F2 | 6.18 ± 0.12a | 59.24 ± 3.55c |
| R8 | Heihe 43 | CK | 6.37 ± 0.16a | 99.79 ± 5.55a |
| | | F1+B | 6.49 ± 0.18a | 97.65 ± 5.84a |
| | | F1 | 6.48 ± 0.15a | 91.37 ± 16.33a |
| | | F2+B | 6.52 ± 0.13a | 97.95 ± 5.52a |
| | | F2 | 6.47 ± 0.13a | 90.44 ± 4.32a |
| | Keshan 1 | CK | 6.22 ± 0.11b | 94.55 ± 4.35a |
| | | F1+B | 6.58 ± 0.12ab | 85.61 ± 3.15b |
| | | F1 | 6.37 ± 0.16ab | 71.44 ± 2.95c |
| | | F2+B | 6.66 ± 0.16a | 87.55 ± 4.3ab |
| | | F2 | 6.41 ± 0.17ab | 83.54 ± 6.6ab |

**Note:**
Different lowercase letters indicate significant differences between the treatments ($P < 0.05$).

treatments (F1+B, F2+B) resulted in higher soil pH compared to the reduced fertilization treatments (F1, F2) and conventional fertilization (CK), but the differences were not statistically significant. The soil pH values across different treatments and stages fluctuated between 6.1 and 6.8, with the optimal pH for crop growth being around 6.5. The biochar treatments (F1+B, F2+B) approached this optimal value most closely, suggesting that biochar significantly influences soil pH and can mitigate soil acidification caused by the long-term use of chemical fertilizers. As seen from Tables 4 to 5, in 2022, there was a slight increase in pH values across all treatments compared to 2021, with no significant changes, following the same trend as the previous year.

Tables 4 and 5 indicate that soil electrical conductivity varied significantly among different treatments. In 2021, the conventional fertilization treatment (CK) consistently showed the highest soil electrical conductivity across different years and varieties, and in 2022, the electrical conductivity for all treatments was slightly higher than in 2021. This suggests that the introduction of biochar and chemical fertilizers introduced a large amount of soluble salts to the soil, increasing the EC value. Additionally, the soil EC values in the biochar and chemical fertilizer treatments gradually increased compared to the reduced fertilization treatments, suggesting that the introduction of biochar facilitated frequent ion exchange between crop roots and soil solution.

As shown in Fig. 2, in the Heihe 43 soybean fields in 2021, soil cation exchange capacity (CEC) gradually decreased as the growth stages progressed. In 2022, the biochar and chemical fertilizer treatments (F1+B, F2+B) showed an increase in CEC compared to 2021. During the R8 stage in 2021, the biochar and chemical fertilizer treatments (F1+B, F2+B) significantly increased CEC by 18.8% and 15.6%, respectively, compared to the reduced fertilization treatments (F1, F2) ($P < 0.05$). Compared to the CK treatment, there was no significant increase in CEC in the biochar treatments.

As depicted in Fig. 3, in the Keshan 1 soybean fields in 2022, soil CEC slightly decreased over the growth stages. During the R4 and R6 stages, the biochar and chemical fertilizer treatments (F1+B, F2+B) showed a significant increase of 26–32% in CEC compared to the reduced fertilization treatments (F1, F2) ($P < 0.05$). At the R8 stage, there was a significant difference in CEC between the biochar and chemical fertilizer treatments (F1+B, F2+B) and the reduced fertilization treatments (F1, F2). Throughout the growing stages in 2022, the CEC continuously decreased in the CK, F1, and F2 treatments, while the biochar and chemical fertilizer treatments (F1+B, F2+B) showed a significant improvement in soil CEC. This indicates that biochar can effectively enhance soil CEC. Compared to the F1 and F2 treatments, the CEC in the F1+B and F2+B treatments significantly increased by 9.6–44.3% ($P < 0.05$).

Zeta potential measurements were conducted on soil samples from the maturity stage of various treatments in 2022. As shown in Fig. 4, the zeta potential values of the chernozem colloids were negative across different varieties, showing a consistent trend. With the increase in biochar addition, the zeta potential shifted towards more negative values, indicating an increase in the negative charge on the soil surface after the addition of biochar. The absolute values of the zeta potential for the F2+B and F1+B treatments were significantly greater than those of other treatments, suggesting that the application of

**Table 5 Effects of biochar and fertilizer on soil pH and EC in soybean planting area in 2022.**

| Periods | Varieties | Treatments | Soil pH | EC (ms/m) |
|---|---|---|---|---|
| R2 | Heihe 43 | CK | 6.19 ± 0.25c | 71.45 ± 1.15a |
| | | F1+B | 6.41 ± 0.35b | 63.85 ± 2.45b |
| | | F1 | 6.35 ± 0.25bc | 56.90 ± 0.70c |
| | | F2+B | 6.52 ± 0.30a | 65.75 ± 4.05a |
| | | F2 | 6.20 ± 0.19c | 56.20 ± 6.10b |
| | Keshan 1 | CK | 6.28 ± 0.18a | 78.32 ± 2.79a |
| | | F1+B | 6.44 ± 0.14a | 71.55 ± 2.15b |
| | | F1 | 6.38 ± 0.19a | 66.45 ± 4.75a |
| | | F2+B | 6.44 ± 0.16a | 75.35 ± 4.65a |
| | | F2 | 6.31 ± 0.20a | 69.63 ± 2.60b |
| R4 | Heihe 43 | CK | 6.25 ± 0.35c | 69.95 ± 2.45a |
| | | F1+B | 6.46 ± 0.43b | 66.15 ± 2.25a |
| | | F1 | 6.39 ± 0.16bc | 62.20 ± 0.74b |
| | | F2+B | 6.59 ± 0.33a | 68.75 ± 1.55a |
| | | F2 | 6.37 ± 0.28c | 64.20 ± 3.33ab |
| | Keshan 1 | CK | 6.28 ± 0.19a | 77.21 ± 2.66a |
| | | F1+B | 6.34 ± 0.14a | 74.25 ± 3.44a |
| | | F1 | 6.18 ± 0.17a | 65.15 ± 3.75c |
| | | F2+B | 6.34 ± 0.16a | 70.15 ± 2.66b |
| | | F2 | 6.21 ± 0.10a | 68.60 ± 2.13bc |
| R6 | Heihe 43 | CK | 6.22 ± 0.15c | 117.6 ± 17.20a |
| | | F1+B | 6.54 ± 0.12ab | 65.8 ± 5.40b |
| | | F1 | 6.35 ± 0.14c | 60.55 ± 4.75b |
| | | F2+B | 6.63 ± 0.15a | 93.65 ± 5.1a |
| | | F2 | 6.51 ± 0.17b | 56.7 ± 8.50b |
| | Keshan 1 | CK | 6.33 ± 0.12a | 83.6 ± 3.8a |
| | | F1+B | 6.37 ± 0.12a | 68.9 ± 5.0ab |
| | | F1 | 6.18 ± 0.21a | 68.8 ± 6.45ab |
| | | F2+B | 6.37 ± 0.11a | 75.4 ± 4.25ab |
| | | F2 | 6.13 ± 0.11a | 59.3 ± 3.05b |
| R8 | Heihe 43 | CK | 6.41 ± 0.15b | 152.3 ± 26.7a |
| | | F1+B | 6.54 ± 0.15a | 97.15 ± 5.6b |
| | | F1 | 6.44 ± 0.16ab | 95.7 ± 5.8b |
| | | F2+B | 6.62 ± 0.13a | 121.6 ± 9.0a |
| | | F2 | 6.34 ± 0.13b | 71.05 ± 5.8b |
| | Keshan 1 | CK | 6.28 ± 0.18b | 91.45 ± 4.35a |
| | | F1+B | 6.49 ± 0.06a | 82.45 ± 2.05ab |
| | | F1 | 6.31 ± 0.02b | 69.35 ± 2.95b |
| | | F2+B | 6.54 ± 0.05a | 85.6 ± 4.3ab |
| | | F2 | 6.33 ± 0.06a | 82.0 ± 6.6ab |

**Note:**
Different lowercase letters indicate significant differences between the treatments ($P < 0.05$).

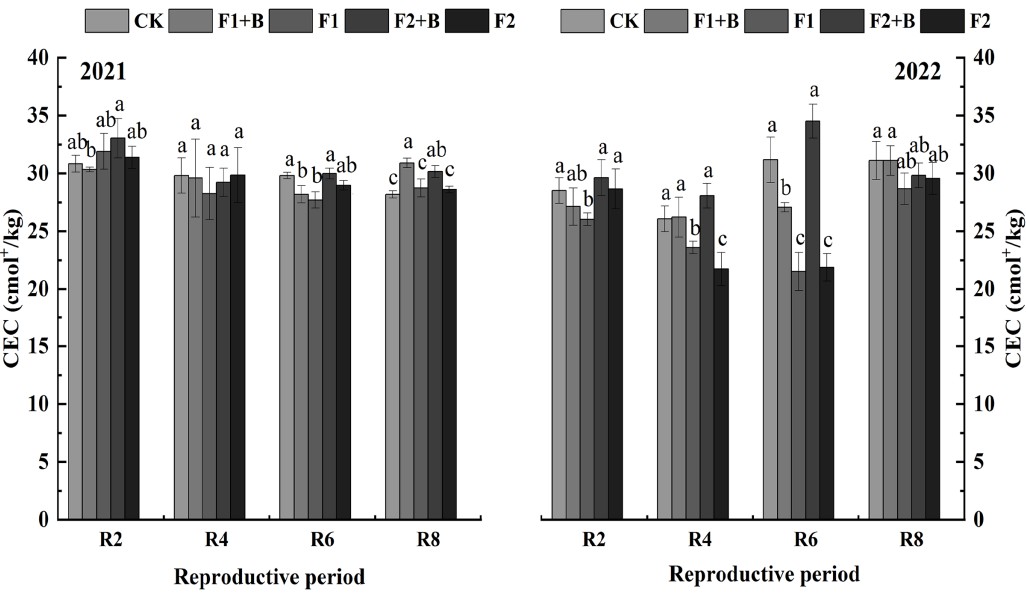

**Figure 2 Effects of combined application of biochar and chemical fertilizer on soil CEC in soybean planting area of Heihe 43.** In the R8 phase of 2021, the combined application of biochar and chemical fertilizer (F1+B, F2+B) significantly increased by 18.8% and 15.6% compared with the weight loss treatment (F1, F2). Different lowercase letters indicate significant differences between the treatments ($P < 0.05$).

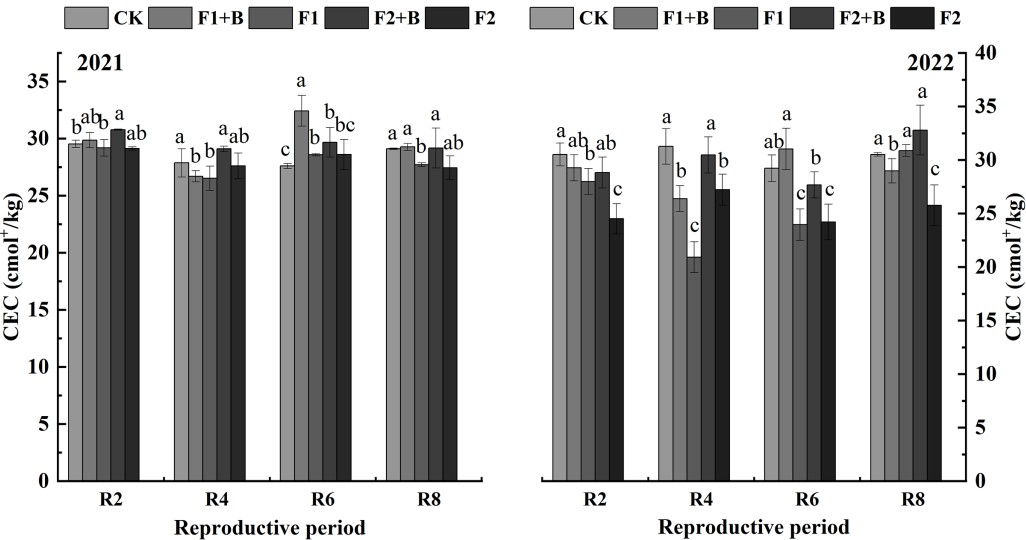

**Figure 3 Effects of combined application of biochar and chemical fertilizer on soil CEC in Keshan No. 1 soybean planting area.** There was a significant difference between R8 biochar and fertilizer treatment (F1+B, F2+B) and weight loss treatment (F1, F2). Different lowercase letters indicate significant differences between the treatments ($P < 0.05$).

biochar enhanced the soil colloids' ability to adsorb negative ions, thereby improving the stability of the soil colloid solution. Additionally, as the growth stages progressed, the zeta potential of the soil colloids showed a decreasing and stabilizing trend.

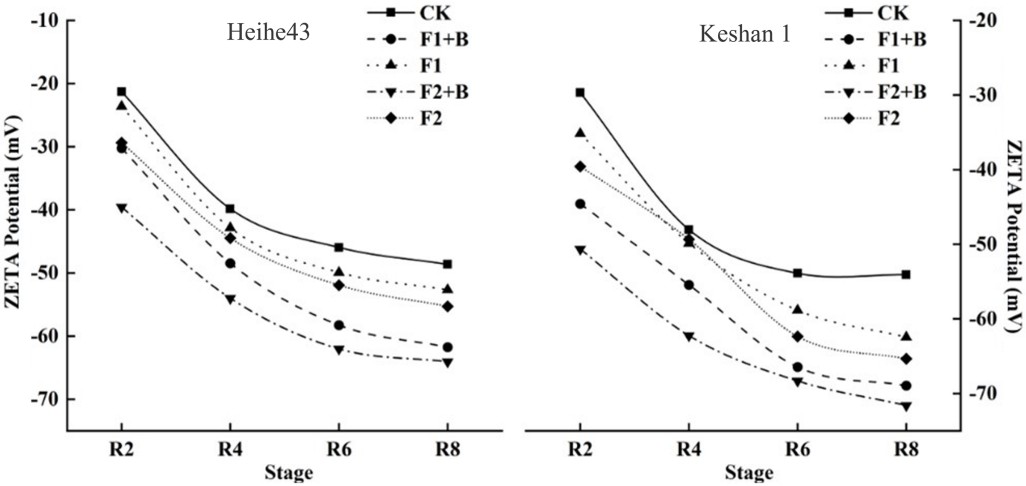

**Figure 4 Effects of combined application of biochar and chemical fertilizer on soil zeta potential in soybean planting area.** The absolute value of zeta potential of F2+B treatment and F1+B treatment was much higher than that of other treatments, indicating that the application of biochar improved the ability of soil colloid to absorb negative ions and improved the stability of soil colloid solution.

**Table 6 Effects of fertilizer and biochar application on surface electrochemical properties of soybean varieties.**

| Varieties | Treatments | SCN cmol/kg | SSA m$^2$/g | σ0 c/m$^2$ | φ0 mV | E0 −10$^8$V/m |
|---|---|---|---|---|---|---|
| Heihe 43 | CK | 16.5 ± 0.07c | 35.7 ± 2.49d | 0.33 ± 0.02a | −118.11 ± 1.22c | 4.77 ± 0.34ab |
| | F1+B | 20.5 ± 0.10b | 64.7 ± 3.62b | 0.32 ± 0.02a | −108.95 ± 1.09a | 5.51 ± 0.23b |
| | F1 | 14.9 ± 0.07c | 43.02 ± 0.55c | 0.31 ± 0.0la | −103.84 ± 0.52b | 4.73 ± 0.08ab |
| | F2+B | 22.9 ± 0.25a | 72.62 ± 5.95a | 0.34 ± 0.03a | −107.51 ± 2.16a | 5.31 ± 0.38b |
| | F2 | 14.5 ± 0.09d | 43.87 ± 2.23c | 0.31 ± 0.02a | −105.54 ± 1.27a | 4.23 ± 0.25b |
| Keshan 1 | CK | 19.5 ± 1.02c | 29.1 ± 2.84d | 0.44 ± 0.05a | −119.87 ± 2.31a | 6.25 ± 0.67a |
| | F1+B | 21.2 ± 0.33b | 57.8 ± 5.90b | 0.42 ± 0.05ab | −121.69 ± 4.23a | 6.67 ± 0.74ab |
| | F1 | 18.3 ± 1.01d | 44.60 ± 3.57c | 0.36 ± 0.04b | −111.29 ± 1.39b | 6.15 ± 0.7lab |
| | F2+B | 23.86 ± 0.21a | 67.1 ± 3.79a | 0.43 ± 0.05ab | −117.23 ± 4.36a | 6.85 ± 0.30b |
| | F2 | 13.6 ± 0.05e | 47.16 ± 3.80c | 0.34 ± 0.02c | −112.60 ± 2.64b | 6.04 ± 0.60b |

**Note:**
Different lowercase letters indicate significant differences between the treatments ($P < 0.05$).

Measurements were carried out on soil samples from the maturity stage of various treatments in 2022. According to Table 6, the soil surface charge number (SCN) for the soybean variety Keshan 1 was slightly higher than that for the Heihe 43 variety. Furthermore, as the level of fertilizer reduction increased in the soybean fields, the soil SCN decreased, while the addition of biochar helped to increase the SCN. For both soybean varieties, Heihe 43 and Keshan 1, the biochar and chemical fertilizer treatments (F1+B, F2+B) significantly increased the soil surface charge number by 33.3%, 51.7% and 16.4%, 73.5% respectively compared to the reduced fertilizer treatments (F1, F2) ($P < 0.05$). Soil specific surface area (SSA), surface charge density (σ0), surface potential (φ0), and surface electric field strength (E0) all followed the same pattern, showing that the biochar and

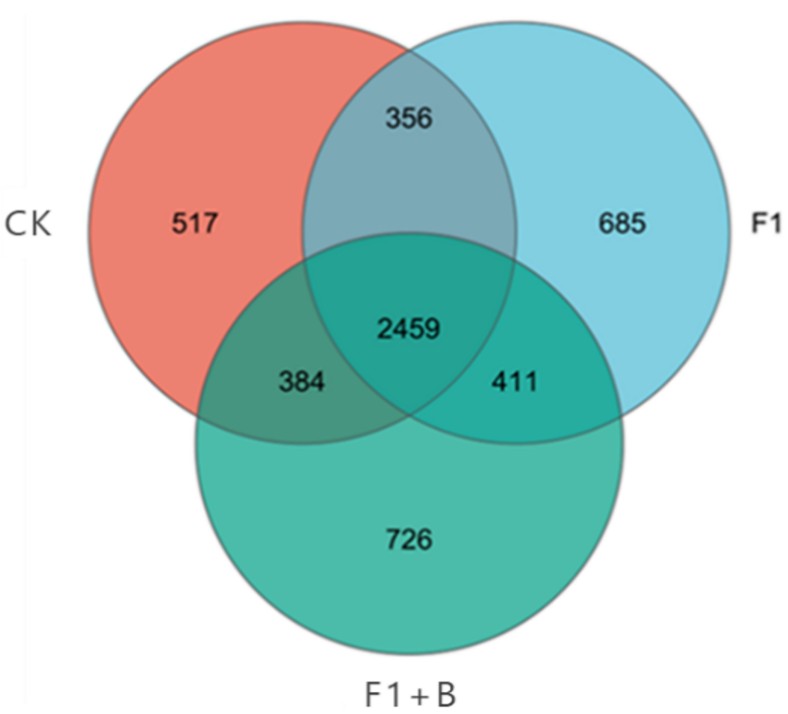

**Figure 5 Venn plot of the number of soil bacteria samples in soybean planting area.** The specific performance of each treatment is F1+B > F1 > CK.

chemical fertilizer treatments (F1+B, F2+B) compared to the reduced fertilizer treatments (F1, F2) enhanced the soil's surface electrochemical properties.

## Impact of combined biochar and chemical fertilizer application on soil bacteria in soybean fields

As shown in Fig. 5, the three treatments had a total of 9,305 OTUs, while the three treatments had 2,459 identical OTUs. Among the unique features, the performance of each treatment was as follows: F1+B > F1 > CK, with the F1+B treatment having the highest number of unique OTUs, accounting for 34.23% of the total number. Overall, the treatments with combined biochar and chemical fertilizer application had a higher total number of OTUs and unique features; the effect of enhancing soil bacterial diversity was evident in the treatments combining biochar and chemical fertilizers.

From Fig. 6, it is evident that in the bacterial community abundance of different soil treatments, the dominant bacterial phyla include Actinobacteria, Acidobacteria, Chloroflexi, Proteobacteria, Gemmatimonadetes, Verrucomicrobia, Firmicutes, Myxococcota, Planctomycetes, and Bacteroidetes. Among these, Actinobacteria, Acidobacteria, Chloroflexi, and Proteobacteria are the predominant phyla within the bacterial community structure. Comparing different fertilization treatments, the F1+B treatment exhibited the highest abundance ratios of Actinobacteria to Proteobacteria. In contrast to the F1+B treatment, the CK and F1 treatments showed the lowest proportions of Actinobacteria and a higher proportion of Acidobacteria, indicating that the combined application of biochar and chemical fertilizers can increase the proportion of

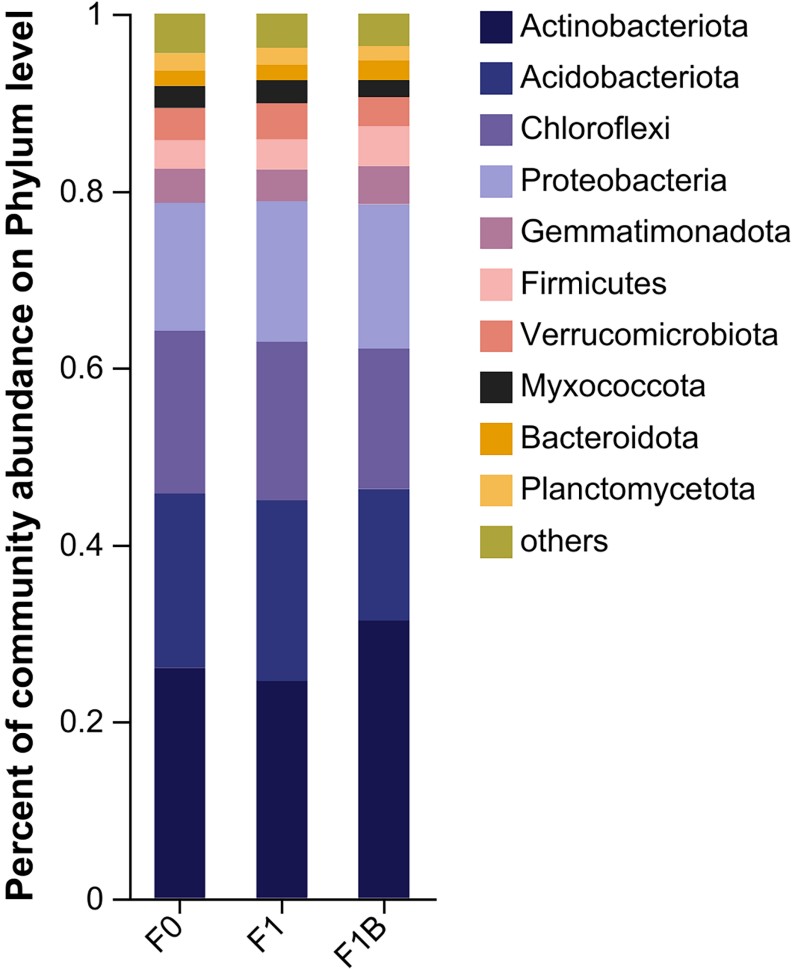

**Figure 6  Soil bacterial community composition in soybean planting area.**

Actinobacteria in the soil. By analyzing the abundance of the dominant phyla across treatments, it was found the proportions of Acidobacteria and Chloroflexi were reduced. Overall, by comparing the bacterial community distribution at the phylum level across different treatments, it is clear that the combined biochar and fertilizer treatment (F1+B) significantly enhances the abundance of Actinobacteria.

Figure 7 displays the F1+B treatment increased the relative abundance of dominant bacterial groups, and the similarity in phylum abundance between the biochar-treated (F1 +B) and non-biochar-treated (CK, F1) samples was low. This indicates that in the soil bacterial community structure, the combined application of biochar and chemical fertilizers can increase the relative abundance ratios of dominant bacterial genera. The combined biochar and fertilizer treatment (F1+B) has a significant impact on the relative microbial abundance in the soil bacterial community.

Table 7 visually reflects the differences in bacterial Alpha diversity indices among different fertilization treatments. The Chao1, Ace, and Shannon indices for the soil F1+B treatment are higher than those for the CK and F1 treatments, but without significant

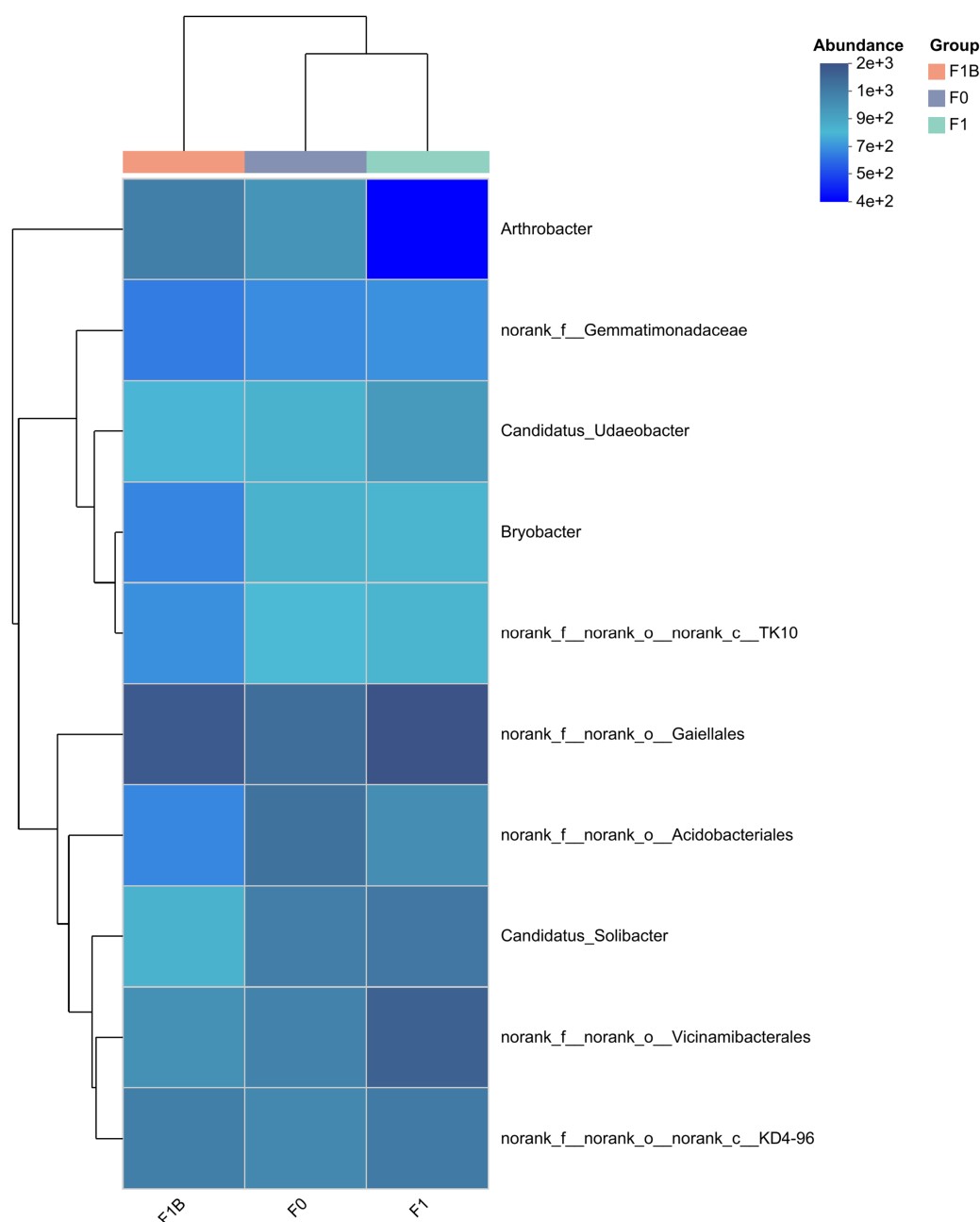

**Figure 7 Clustering of species abundance of soil bacterial communities in soybean planting area.**

**Table 7 Alpha diversity index of soil samples.**

| Treatments | Index of abundance | | Diversity index | |
|---|---|---|---|---|
| | ACE index | Chao1 index | Shannon index | Simpson index |
| CK | 3,458.21 ± 174.77a | 3,395.77 ± 179.13a | 0.01 ± 0.01a | 2,553.33 ± 29.09a |
| F1 | 3,430.11 ± 203.2a | 3,315.44 ± 152.7a | 0.01 ± 0.01a | 2,631.33 ± 133.36a |
| F1+B | 3,538.34 ± 44.86a | 3,435.21 ± 126.2a | 0.01 ± 0.01a | 2,658.0 ± 38.12a |

**Note:**
Different lowercase letters indicate significant differences between the treatments ($P < 0.05$).

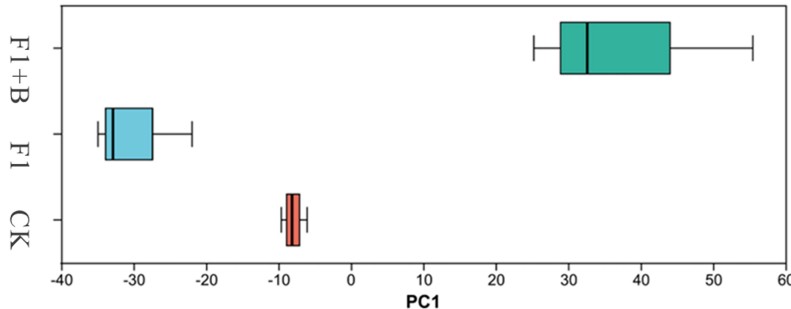

**Figure 8 PCA analysis of soil bacteria in soybean planting area.** The Chao1, Ace and Shannon indices of soil F1+B treatment were higher than those of CK and F1 treatment, but there was no significant change.

changes, indicating that the combined application of biochar and chemical fertilizers could slightly increase the bacterial diversity.

As shown in Fig. 8, PCA analysis showed that there was little similarity between the bacterial communities of soybean at phylum level, with significant differences observed between the F1+B treatment and the CK and F1 treatments. This suggests that the addition of biochar can alter the structure of the bacterial microbial community. The results indicate that the combined application of biochar and chemical fertilizers has altered the structure of the soil bacterial microbial community.

## Impact of combined biochar and chemical fertilizer application on soil fungi in soybean fields

As depicted in Fig. 9, the three treatments shared a total of 2,132 fungal community features, with 489 OTUs common across all three samples. Among the unique OTUs, the specific performance of each treatment was as follows: F1+B > CK > F1, with the F1+B treatment having the highest number of unique OTUs, accounting for 36.77% of the total number. Overall, the treatments with combined biochar and chemical fertilizer application exhibited a higher total number of OTUs and unique OTUs for both bacteria and fungi; the effect of enhancing soil fungal diversity was clearly significant in the treatments combining biochar and chemical fertilizers.

As shown in Fig. 10, the distribution of soil fungal communities in different treatments is primarily dominated by the phyla Ascomycota, Basidiomycota, and Mortierellomycota. Among these treatments, the abundance of Ascomycota in the CK treatment is significantly lower than in the F1+B and F1 treatments, but the proportion of Basidiomycota is higher compared to the other treatments. Additionally, the abundance of Mortierellomycota is significantly increased in the F1 and F1+B treatments.Overall, by comparing the distribution of fungal communities at the phylum level under different treatments, it is evident that the combined application of biochar and fertilizer (F1+B) significantly increased the abundance of Ascomycota and Mortierellomycota fungal populations.

In the soil fungal abundance heatmap, as illustrated in Fig. 11, eight out of the top ten microbial by total abundance at the taxonomic level belong to the phylum Ascomycota,

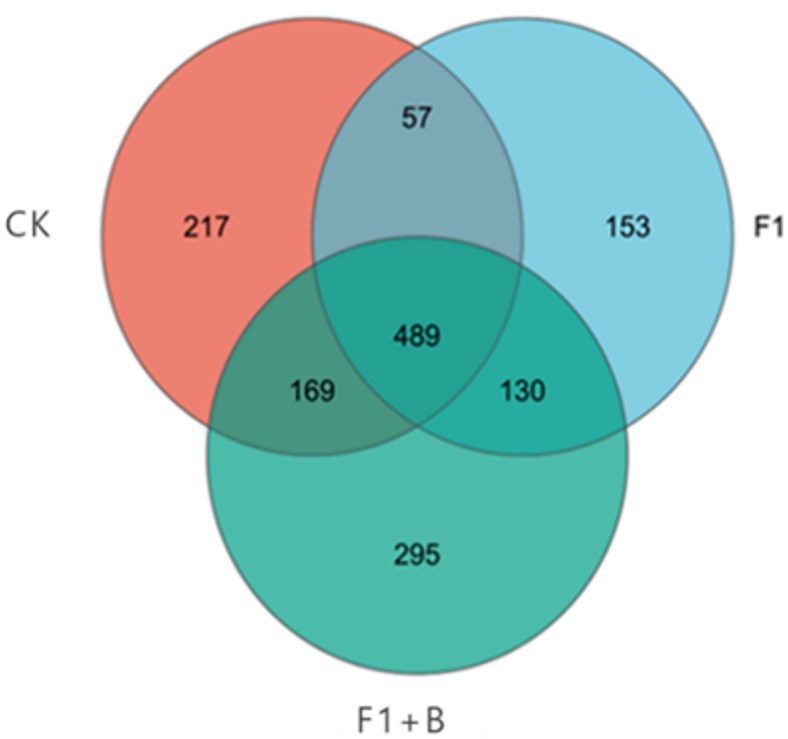

**Figure 9 Venn plot of the number of soil fungal samples in soybean planting area.** The specific performance of each treatment is F1+B > CK > F1, F1+B treatment has the largest number of OUT samples.

which has the largest representation. The microbia abundance distribution of F1+B and F1 treatments is relatively similar. Compared with CK treatment, the abundance proportion of Basidiomycota microbia decreased to Neobulgarias under F1+B and F1 treatments, while the abundance proportion of Mortierellomycota microbia increased to Mortierella.

In Table 8, differences in fungal Alpha diversity indices among various fertilization treatments are presented. The F1+B treatment has the highest Chao1, Ace, and Shannon indices, and these indices are significantly higher in the F1+B treatment compared to the CK and F1 treatments; the Simpson index in the F1+B treatment is significantly lower compared to the CK treatment. Overall, these results indicate that the combined application of biochar and chemical fertilizers increases the fungal community diversity.

As shown in Fig. 12, PCA analysis showed that there was little similarity among soybean fungal communities at phylum level, with significant differences observed between the F1+B treatment and the CK and F1 treatments. This indicates that the addition of biochar can alter the structure of the fungal microbial community. This suggests that the combined application of biochar and chemical fertilizers has changed the structure of the soil fungal microbial community.

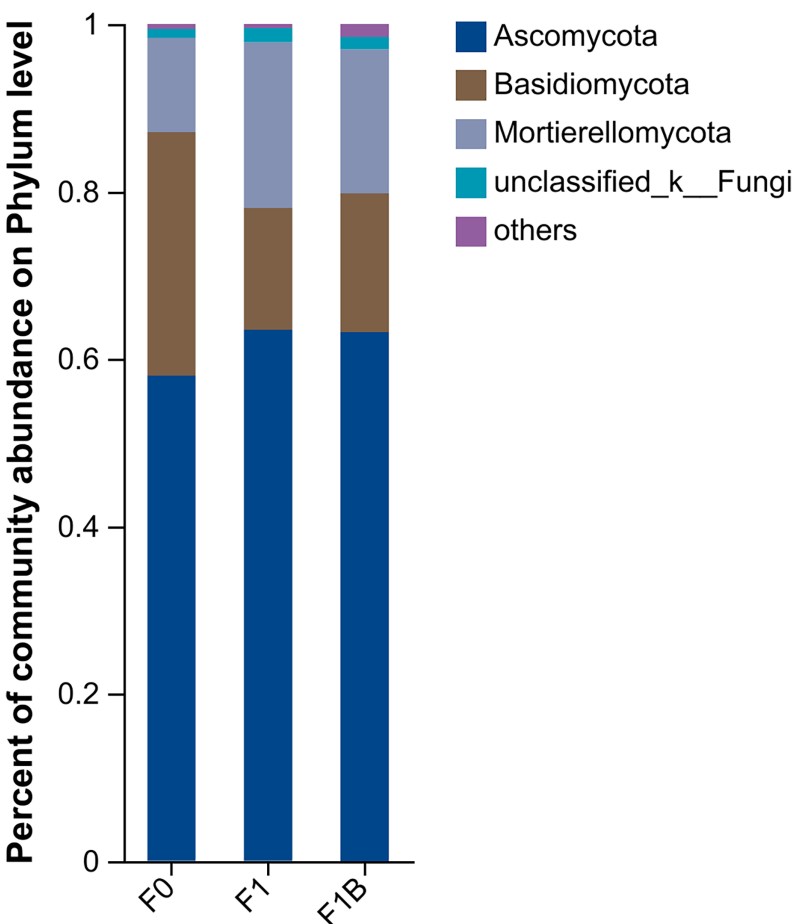

**Figure 10** **Soil fungal community composition in soybean planting area.**

## DISCUSSION

### Impact of combined biochar and chemical fertilizer application on soil chemical properties

Numerous studies indicate that prolonged use of chemical fertilizers leads to a continuous decline in soil pH (*Ruan et al., 2023*), an increase in electrical conductivity (*Faloye et al., 2022*), an excessive accumulation of soluble salts (*Ma, 2021*), and a degradation of soil quality, resulting in reduced organic matter content and nutrient imbalance in the soil. Extensive research has confirmed that the co-application of biochar and chemical fertilizers can effectively buffer changes in soil pH and electrical conductivity, thereby enhancing soil quality (*Zhang et al., 2023*; *Guo et al., 2023*). In this study, analysis of the pH variations in soybean fields treated differently over 2 years revealed that soils without biochar application (F1, F2) maintained a pH range of 6.16–6.3, significantly lower than that of soils treated with both biochar and chemical fertilizers (F1+B, F2+B), which is not conducive to the healthy growth of crop roots, highlighting that fertilization rate is a major factor affecting soil pH. The combined biochar and fertilizer treatments were able to restore soil pH to the optimal range for crop growth, closely related to the alkaline

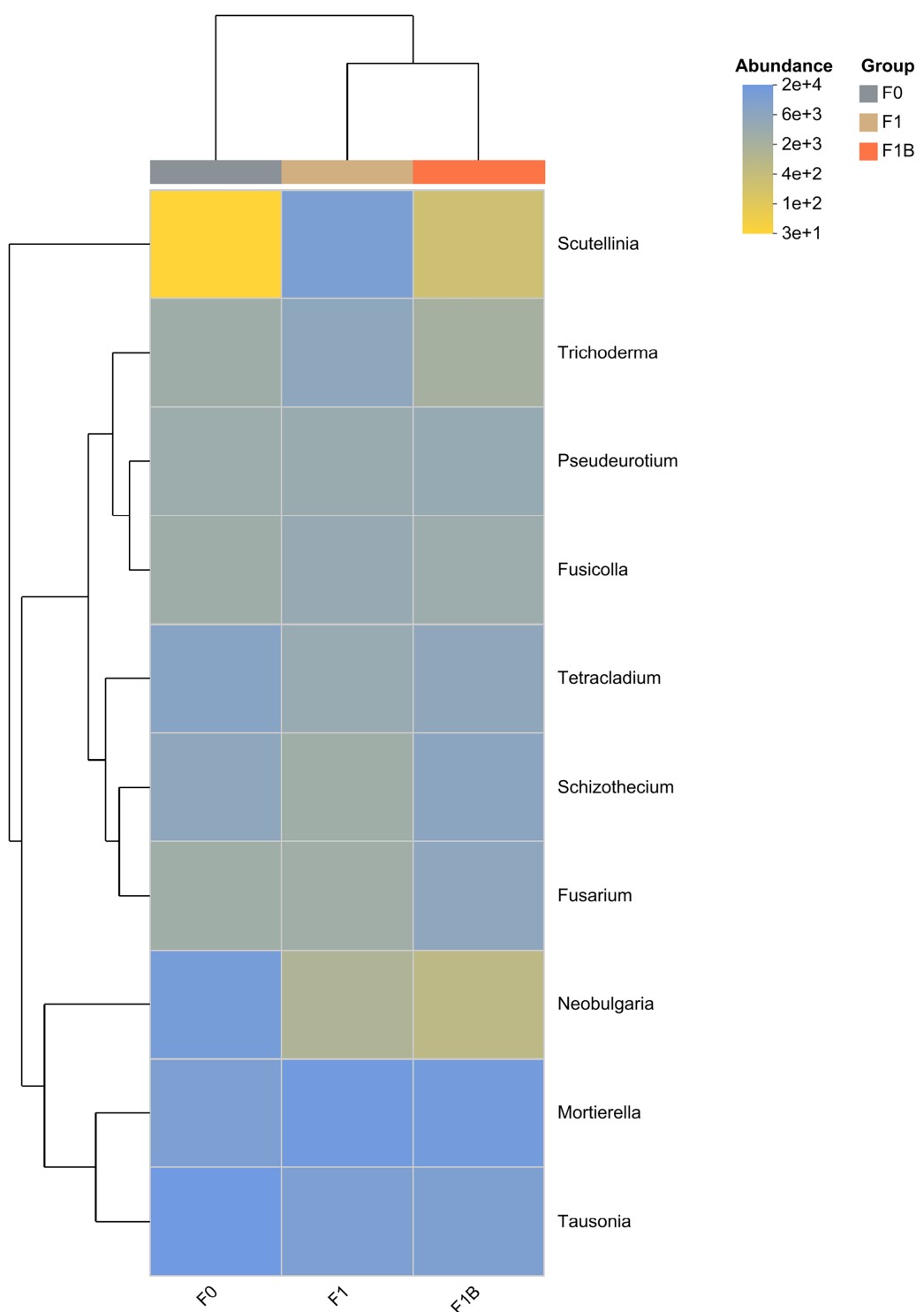

**Figure 11 Clustering of species abundance of soil fungal communities in soybean planting area.**

 

**Table 8 Alpha diversity index of soil samples.**

| Treatments | Index of abundance | | Diversity index | |
|---|---|---|---|---|
| | ACE index | Chao1 index | Shannon index | Simpson index |
| CK | 617.51 ± 28.11b | 608.77 ± 27.32b | 3.31 ± 0.29c | 0.11 ± 0.04a |
| F1 | 572.52 ± 39.25c | 573.42 ± 36.4c | 3.63 ± 0.12b | 0.06 ± 0.01ab |
| F1+B | 733.9 ± 45.41a | 728.73 ± 43.49a | 4.08 ± 0.1a | 0.04 ± 0.01b |

**Note:**
  Different lowercase letters indicate significant differences between the treatments ($P < 0.05$).

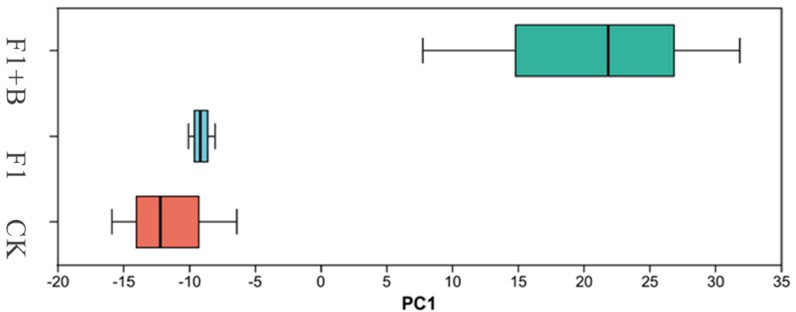

**Figure 12 PCA analysis of soil fungi in soybean planting area.** Soybean F1+B treatment was significantly different from CK and F1 treatment.

functional groups (–COO–, –O–, –OH) inherent in biochar. Additionally, applying biochar changes the transformation and proton dynamics of nitrogen in the soil, thereby affecting soil pH (*Fidel et al., 2017*). Studies have shown that biochar application significantly accelerates the mineralization of organic nitrogen in soil, a process that essentially consumes protons, thereby raising soil pH and enhancing root absorption of readily available nutrients (*Li, 2022*; *Yu et al., 2020*). This study also found that long-term use of chemical fertilizers significantly increases soil electrical conductivity, which is related to high soil bulk density, increased soil compaction, salt accumulation at the soil surface, lack of adequate rainfall leaching, leading to a rapid increase in total soil salinity and accelerated secondary salinization (*Shi et al., 2022*). In this study, soils treated with biochar and fertilizers (F1+B, F2+B) showed significantly reduced electrical conductivity and increased cation exchange capacity (CEC). This is possibly due to biochar's rich porosity and surface area enhancing soil porosity and aeration, speeding up the leaching of surface salts. And its oxidized surface functional groups increasing cation adsorption capacity, thereby increasing cation exchange and reducing nutrient leaching (*Lago et al., 2021*; *Domingues et al., 2020*). This is consistent with Igalavithana's research, which found that biochar typically has higher CEC than most agricultural soils due to its high soluble salt and oxygen-containing alkaline functional group content, which enhances the effectiveness of soluble nutrients like $NO_3^-$, $K^+$, and $Ca^{2+}$ (*Igalavithana et al., 2018*).

   Soil surface electrochemical properties are determined by the balance and exchange of ions in the soil solution, and the interactions between soil particles profoundly affect soil aggregate stability (*Liu et al., 2022*), ion exchange, and the absorption and transport of

nutrients (*Sabda, Junun & Makruf, 2022*). The zeta potential characterizes the stability of soil colloids; smaller dispersed particles have higher absolute potentials, indicating stronger stability of the soil colloid solution or aggregates. In this study, the absolute zeta potentials of soils treated with biochar and chemical fertilizers (F1+B, F2+B) were higher than other treatments, likely influenced by the ion strength. With the addition of biochar, soil pH increases, ion strength decreases, the concentration of coexisting cations in the suspension decreases, and the soil colloid double layer structure is released, raising the absolute zeta potential (*Hong et al., 2019*). As the amount of biochar increased, the soil's zeta potential shifted towards more negative values due to the deprotonation of oxygen-containing functional groups ($-COO-$ and $-OH$) on the biochar surface, increasing the surface negative charge and thus the absolute zeta potential (*Cui et al., 2022*). This shift towards negative values continues throughout the growth period, possibly because as the season progresses, a significant amount of ammonium ions converted from organic forms in the soil interact dependently with the functional groups in the biochar and the numerous $H^+$ ions in the soil solution (*Liu et al., 2018*). The vigorous reproductive growth of crops intensifies soil solution reactions, consuming large amounts of protons and causing an increase in pH. The surface electrochemical properties of the soil are a major factor affecting the interactions between soil particles, aggregate stability, and erosion resistance. In the treatments combining biochar and chemical fertilizers (F1+B, F2+B), the quantity of surface charges, specific surface area, and surface charge density of the soil were significantly increased. This is primarily due to the rich surface area and porosity of the biochar, which effectively increases the soil's specific surface area and adsorption of water molecules and ions, thereby increasing the stability of the dispersion and enhancing the quantity and density of soil surface charges (*Zhao et al., 2022*). Yang Caidi's study indicated that returning straw char to fields significantly improves soil surface charge quantity and CEC, with increases of 67.4% and 50.9% respectively compared to returning straw directly (*Yang, Liu & Lu, 2023*). This study also noted that under the treatments combining biochar and fertilizers, soil organic matter content and soil surface charge density were enhanced, suggesting that soil organic matter plays a crucial role in the adsorption of soil nutrient ions. The analysis may be due to the addition of biochar increasing the soil's specific surface area, which in turn increases the unit soil charge density as more biochar is added (*Yang et al., 2022*). Scientific research has found that soil surface electrochemical properties are closely related to organic carbon content, which can promote the formation of organic-inorganic complexes in the soil. And the high surface area charge density and increased cation adsorption sites enhance the quantity of soil surface charges (*Wang et al., 2017*; *Zhao, 2018*). Studies have shown that soil organic carbon content is significantly positively correlated with soil specific surface area (*Kimetu & Lehmann, 2010*). Biochar's high specific surface area not only improves the soil's specific surface area but also provides numerous attachment sites, significantly enhancing the adhesion rate of soil organic carbon, thereby increasing soil organic carbon content. Additionally, after biochar is applied into soil, functional groups in organic carbon change, and active groups such as carboxyl group, hydroxyl group and aldehyde group increase (*Liyanage et al., 2021*), which

can dissociate H$^+$ and increase the negative charge on soil surface. This is also another reason for the improvement of soil surface electrochemical strength.

## Impact of combined biochar and chemical fertilizer application on soil microbial community

Soil microbes are a crucial component of the soil ecosystem, playing important roles in decomposing organic matter, improving soil structure and fertility, ecological regulation, and enhancing crop growth (*Chen et al., 2021*). Extensive research indicates that the scientific application of chemical fertilizers can effectively improve the soil's ecological environment, increase the diversity of soil microbes, and foster synergistic interactions between soil microbes and plants, thereby enhancing the self-regulating and protective capacities of farmland soil (*Yang et al., 2022*). Using high-throughput sequencing technology, this study analyzed the microbial characteristics of soybean field soils treated with biochar and chemical fertilizers, exploring the diversity and relative abundance of microbial phyla under these conditions. Under each treatment in this study, Actinobacteria and Protepbacteria were the dominant groups in the composition of bacterial communities. In natural settings, Proteobacteria play a significant ecological role by decomposing soil organic matter and providing nutrients to crops, although they also include pathogens that can threaten crop health (*De Tender et al., 2016*). Actinobacteria can decompose organic matter as well, producing a large amount of active substances, secreting organic acids, and extracellular polysaccharides that are crucial for forming soil aggregates and improving soil aeration and water retention (*Warnock et al., 2007*). They also synthesize amino acids and polysaccharides that enhance crop resilience and immunity, mitigating damage caused by environmental stress and thus promoting crop growth (*Guan et al., 2023*). Additionally, in the F1+B treatment, Ascomycota, Basidiomycota, and Mortierellomycota were the dominant fungal phyla in the soil microbial community, with their relative abundance closely related to soil nutrient content and material cyclin (*Wu, 2022*). These dominant fungal and bacterial communities compared to reduced fertilizer treatments, enhancing the connection between leguminous crops and symbiotic nitrogen-fixing microbial groups in the soil and effectively increasing soil nitrogen availability, which is significant for nitrogen fixation in soybean nodules and crop growth (*Zhang, 2022*). Moreover, dominant microbial groups can absorb harmful elements from the soil, reducing soil pollution, and promote the exchange of gases and energy release between microbial communities and the soil (*Zhang et al., 2018*), enhancing the crop's resilience (*Luo et al., 2019*).

In the analysis of soil microbial community Alpha and Beta diversity, the impact of the F1+B treatment on soil fungal communities was more pronounced, with higher fungal microbia richness and evenness under the F1+B treatment, as reflected in the higher Chao1 and Ace indices, consistent with previous research findings (*Li et al., 2020*). This may be due to the fact that by improving soil structure, biochar enhances the resilience of fungal communities. In the case of the same microbia richness, it indicates that biochar provides a suitable environment for the growth of soil fungal community.

## CONCLUSIONS

During the soybean growing season, the combined application of biochar and chemical fertilizers can increase soil electrical conductivity and cation exchange capacity, enhance the absolute value of soil zeta potential, stabilize the surface charge density of soil colloids, and improve soil surface potential. Under the treatment of 1 t/hm$^2$ of biochar combined with fertilizers (F1+B), the soil's chemical properties are optimized, thereby ameliorating the decline in soil quality caused by long-term fertilizer use. Under the soybean-corn rotation conditions, the treatment with 1 t/hm$^2$ of biochar combined with fertilizers (F1 +B) enhances the community diversity of soil bacterial and fungal. The results provide a theoretical basis for stable yield increase in soybeans.

### Funding

This study was supported by Scientific and Technological Innovation projects (2023DZD0403106), the Natural Science Foundation Project of Heilongjiang Provincial (LH2022D019), and the Postdoctoral Scientific Research Startup Fund Project of Heilongjiang Provincial (LBH-Q21162). The funders had no role in study design, data collection and analysis, decision to publish, or preparation of the manuscript.

### Grant Disclosures

The following grant information was disclosed by the authors:
Scientific and Technological Innovation: 2023DZD0403106.
Natural Science Foundation Project: LH2022D019.
Postdoctoral Scientific Research Startup Fund Project: LBH-Q21162.

### Competing Interests

The authors declare that they have no competing interests.

### Author Contributions

- Wei Xie performed the experiments, prepared figures and/or tables, and approved the final draft.
- Xingjie Zhong performed the experiments, prepared figures and/or tables, and approved the final draft.
- Yuqing Wang performed the experiments, prepared figures and/or tables, and approved the final draft.
- Siyan Li performed the experiments, prepared figures and/or tables, and approved the final draft.
- Yanhong Zhou conceived and designed the experiments, analyzed the data, prepared figures and/or tables, authored or reviewed drafts of the article, and approved the final draft.
- Chen Wang conceived and designed the experiments, analyzed the data, prepared figures and/or tables, authored or reviewed drafts of the article, software application, and approved the final draft.

## DNA Deposition

The following information was supplied regarding the deposition of DNA sequences:

The sequences are available at GenBank: PRJNA1117087.

## Data Availability

The raw data is available in the Supplemental File.

## Supplemental Information

Supplemental information for this article can be found online at http://dx.doi.org/10.7717/peerj.18172#supplemental-information.

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
