# Peer review of "The impact of combined application of biochar and fertilizer on the biochemical properties of soil in soybean fields"

_PeerJ, doi:10.7717/peerj.18172_

## Round 0.1 · original submission · Major Revisions

Dear Authors,

Reviewers have identified concerns in all sections of your manuscript. You are advised to improve your manuscript and resubmit.

·

Basic reporting

Professional article structure, figures, tables. Raw data shared.

Experimental design

Original primary research within Aims and Scope of the journal.

Validity of the findings

Conclusions are well stated, linked to original research question & limited to supporting results.

Additional comments

In the manuscript “The impact of combined application of biochar and chemical fertilizer on the biochemical properties of soil in soybean fields”, the authors have work hard to execute the current study. Title of the manuscript is interesting and covers the scope of the journal. Nevertheless, there are following issues which need to be addressed by the authors:
• Chemical fertilizer is general term. Please revise the tittle.
• Predictive analysis of microbial functions found that the F1+B treatment increased the abundance of saprophytic fungi, promoting the decomposition of animal and plant remains in the soil, slowing down bacterial community cell processes, inhibiting the growth of pathogens, and enhancing the organic systems and metabolism of bacterial communities. Please mention types of pathogens with more clarity.
• The author conclude that fungal groups were Ascomycota, Basidiomycota, and Mortierellomycota. These are broad fungal groups. On what basis author claimed that fungal groups belong to Ascomycota, Basidiomycota, and Mortierellomycota.
• Actinobacteria, Acidobacteria, Chloroûexi, and Proteobacteria. Please add some plates of mentioned bacteria.
• What is novelty of current research.
• The crude protein content is 38.04%, and the crude fat content is 21.82%. Heihe 43 (high-protein type), also shows a semi-determinate pod-setting habit, plant height around 75 cm, non-branching, with purple flowers, pointed leaves, and grey pubescence. The seeds are round, yellow-coated, with a light yellow hilum, glossy, each seed weighing approximately 20 grams, containing 41.84% crude protein and 18.98% crude fat. Pleas add reference source.
• On what basis the biochar treatment was selected. Please add source. The biochar application rates for the soybean fields were set at 1 t/ha and 2 t/ha, while for maize fields, the rates were 1.5 t/ha and 3 t/ha (as determined from preliminary trials).
• Figure 1 contains other language. Please check the x-axis, y-axis.
• What is contribution of PC1?

·

Basic reporting

no comment

Experimental design

no comment

Validity of the findings

no comment

Additional comments

Authors of the manuscript “The impact of combined application of biochar and chemical fertilizer on the biochemical properties of soil in soybean fields” revealed that the combined application of biochar and chemical fertilizers enhanced soil electrical conductivity and cation exchange capacity, the absolute value of soil ZETA potential, surface charge density of soil colloids, soil surface potential and species richness in soil.
It is an interesting study. However, authors need to address the following issues:
1. Why soybean fields? Please provide justification?
2. Please add the implication of this research.
3. Improve language and grammar of the manuscript. e.g. L23 started with “Conducted at the Heshan Farm Science and Technology Park of Beidahuang Group's 24 93rd Subsidiary in Heilongjiang Province”
4. Please add data in results section of abstract.
5. Please arrange keywords alphabetically.
6. Introduction: Some modifications must be done like many sentences were 4-6 lines long e.g. one sentence started from L60 and ends at L66 (almost 90 words), similarly another sentence was L72-L79 long. Please reduce the number of words in a sentence so it could be understandable for the readers.
7. Methodology: L184 to L187, authors mentioned two plots, one for soybean, and another for maize. But the results were taken only for soybean field. Because the purpose of this study was not to compare both crops, then the author should mention only one plot having the soybean last crop.
8. L188 please provide the preliminary data in supplementary section.
9. Why only soil samples from the R8 stage of the CK, F1+B, and F1 treatments in 2022 were sent for high-throughput sequencing of soil microbial characteristics?
10. Which extraction method was used for DNA extraction? Please provide reference.
11. Results: The presentation of the results should be refined. Add data (values or percentages) to validate the results. Remove comments on relevance, and validity of the findings, and focus on a clear and simple description of the results to highlight. For example see L261, L264 etc.
12. Figure 3: Letters indicating significant differences are incorrectly labelled. Please verify the uniformity of formatting throughout the text.
13. Figures should be in High Resolution. Please improve the quality of Fig. 7 and fig. 11.
14. Table 2: CEC (cmol/kg) unit alignment is not centralized.
15. Table 4: in title please replace corn with maize as maize was used in whole manuscript. Please maintain uniformity.
16. Table 5: add a space in Soil electric(ms/m), (please remove this type of typos from whole manuscript) add full word electric conductivity or use EC.
17. Table 7: Effects of fertilizer and carbon application on surface electrochemical properties of soybean and maize varieties (where is data for maize variety?)
18. Please use word “biochar” instead of carbon in title of table 7.
19. Discussion: There is a lack of mechanistic approach. Improve discussion with the help of logics and further relevant literature
20. Conclusions: The contents of this section are appropriate.
21. References: Satisfactory containing sufficient articles published in the last five years. Check the reference formatting manually. Ensure that the formatting is consistent and it does not have formatting errors.
22. Don’t use et al. in a reference list, please provide full author list.

·

Basic reporting

The study determined the biochar and chemical fertilizers applied in the soybean planting area and their impacts on the soil chemical properties and soil microbial communities. The manuscript is not well written or structured (please see specific comments below). The sentences are too long throughout the manuscript. Please use professional service to improve the language.

The authors didn’t explain what methods were used in the study to process the raw reads and what data (genus level?) were used to do PCA or clustering analysis.

Only very few bacteria or fungi have specific functions. The taxonomic information based on 16S rRNA gene couldn’t be connected to specific functions. Please delete the parts about microbial function predictions, unless functional genes or metagenomes were determined in the study.

The figure descriptions about the microbial community patterns (Figures 6, 7, 10, 11) are not completely correct in the main text. The findings for these figures are not correct as well. Please rewrite these parts. Please show pca1 and pca2 for PCA plot to determine the pattern of treatments.

The manuscript has poor description or discussion about the part of microbial communities, which need to be improved a lot. Don’t mix species level with phylum or genus level throughout the manuscript. In addition to PCA and clustering analysis, I would suggest to use bray-curtis distance analysis, ANOSIM and SIMPER based on the OTU/genus data to determine the similarity/dissimilarity between treatments, and which taxa contribute the most to the dissimilarity. Focus on the discussion of taxa enriched in F1+B compared to F1 or CK.

Please add title for Figure 1 and translate Chinese labels into English.

Figs 7 and 11: Please double check what data was used for clustering analysis. Correct the title correspondingly.


Lines 52-104 and Lines 105-143: Please rewrite the whole part about the traditional fertilization, biochar and the co-application, and their impacts on the soil chemical and physical properties and the soil microbial communities . Put together citations and summarize what the previous studies found, instead of describing individual study findings.

Lines 126, 128, 130, 133. 139, 189, 197-198: Please add related citations.

Lines 191-193: Please explain what components are included for each treatment.

Lines 215-216: Please clarify which soybean planting area was selected to collect microbial community samples.

Line 219: Please provide forward and reverse primers used in this study and the reference.

Lines 224-226: Please provide the software or packages used to process raw reads and the reference. OTUs were produced with a way differed from ASVs. Please clarify it is OTU or ASV determined in the study.

Lines 226-229: Rephrase and break it into small sentences. Please provide database used to annotate sequences and the reference. What data (at the genus level?) was used to do clustering analysis?

Lines 231-232: Please clarify what data was used to do PCA analysis. Change it into ‘PCA analysis was used to determine the relationships between different treatments based on fungal and bacterial species compositions’, if the fungal and bacterial species composition data was used here.

Lines 242-251: Please provide the range of soil pHs and the mean value for each treatment.

Lines 252-253: Please provide the range of soil pHs and the mean value for each year.

Line 251: Change ‘Table 3 to 13’ to ‘Table 5 to 6’.

Lines 257-259: Delete this. Lines 255-257 also include CK and reduced fertilizer treatments.

Lines 259-261: Please provide the range of soil electrical conductivity and the mean value for each treatment.

Lines 293-303: Please indicate if these measurements were different between CK and the other treatments.

Lines 307-308: I would suppose ‘9305 bacterial community features’ should be ‘9305 bacterial community ASVs (or OTUs, dependent on the method used, please see the method part comments on this)’. Please change ‘features’ to ‘ASVs (or OTUs)’ throughout the manuscript.

Line 320: Change ‘ratios of Actinobacteria and Proteobacteria’ to ‘ratios of Actinobacteria to Proteobacteria’.

Line 323-326: Change ‘Analyzing’ to ‘By analyzing’. Delete ‘ratios’ because this sentence described the abundance/proportions instead of ratios. Delete ‘Proteobacteria’ which doesn’t show higher abundance in F1+B treatment, compared to F1.

Line 327: Change ‘genus’ to ‘phylum’.

Line 329: Change ‘dominant bacterial population’ to ‘Actinobacteria’.

Lines 331-339: The clustering analysis is based on the bacterial community compositions at the genus level instead of species or phylum level. Please indicate what specific genera were enriched in F1+B, compared to CK and F1. Rewrite the whole paragraph.

Lines 343-344: Change it to ‘the combined application of biochar and chemical fertilizers could slightly increase the bacterial diversity’.

Lines 345-351: Please use pca1 (horizontal axis) and pca2 (vertical axis) to visualize the pattern of samples. Don’t overstate the results and please delete ‘increasing the differences in microbial community structure’ and ‘facilitating the evolution of the soil bacterial community’.

Lines 352-361: The study didn’t detect any functional genes or metagenome data. The taxonomic information based on 16S rRNA gene couldn’t demonstrate biological functions performed by bacteria. Please delete this whole paragraph.

Lines 365-372: The same problem here as lines 307-313. Change ‘features’ to ‘OTUs’.

Lines 373-381: The whole paragraph didn’t show that F1+B could enhance the abundance of any dominant fungal phyla, compared to CK and F1. Figure 10 is based on the fungal community composition at the phylum level. However, genus level description was mixed here. Please check the whole manuscript about the description of different taxonomic levels. Please rewrite this whole paragraph.

Lines 382-390: Please double check what data (at the genus level?) was used here for clustering analysis.

Lines 387-390: Please explain what is a similar distribution in the abundance of Ascomycot? What is a stable distribution? Fig 11 didn’t show significant difference between F1+B and F1.

Lines 396-397: Change it to ‘increases the fungal community diversity’.

Lines 398-418: The same problems as bacterial parts. Please see comments for lines 345-361.

Lines 432-433, 434-435, 462-465, 484-485, 505-507, 507-510: Add references.

Lines 441-449, 468-473, 480-484, 488-491: These sentences are too long. Please break each into small sentences.

Lines 494-564: Change ‘microbial characteristics’ into ‘microbial communities’.

Line 504: Please explain which figure showed that bacterial organisms were more prevalent than fungi in the treated fields.

Lines 505-516: Proteobacteria didn’t show enrichment in F1+B (Fig 6). Please correct this part of discussion.

Lines 516-520: Ascomycota, Basidiomycota, and Mortierellomycota were not enriched in F1+b compared to F1 and CK. Please correct this part of discussion.

Lines 521-528: Please specify what dominant bacterial and fungal groups are referred here. What is the difference between this part and the above discussion part in lines 503-519?

Lines 528-530: F1+B could increase the alpha diversity of microbial communities. It doesn’t mean that F1+B could increase the dominant groups of microbes. Please correct.

Lines 530-533: The study didn’t detect soil enzyme activity and nitrogen accumulation. Please delete this part.

Lines 534-537: Delete ‘beta diversity’. Only alpha diversity indices were discussed here.

Lines 537-538: Change ‘while’ to ‘by’. Delete ‘also’.

Lines 539-543: Please explain what is the same species richness and how to draw the finding about reducing the similarity in microbial abundance? Delete ‘suppressing pathogenic microbes’, ‘accelerates the interactions between soil fungal communities’ and ‘gradually expanding the fungal community’.

Lines 544-564: Delete.

Lines 572-576: F1+B only increase the alpha diversity of microbial communities instead of dominant groups. Also delete lines 574-576 which are overstated.

Experimental design

The authors didn’t explain what methods were used in the study to process the raw reads and what data (genus level?) were used to do PCA or clustering analysis.

Validity of the findings

The findings about the microbial community part needs to be corrected.

---

## Round 0.2 · Minor Revisions

Dear Authors,

Please improve your manuscript by carefully addressing the issues pointed by Reviewer 3.

·

Basic reporting

Self-contained with relevant results to hypotheses

Experimental design

Research question well defined, relevant & meaningful. It is stated how research fills an identified knowledge gap

Validity of the findings

All underlying data have been provided; they are robust, statistically sound, & controlled

Additional comments

No further comments.

·

Basic reporting

no comment

Experimental design

no comment

Validity of the findings

no comment

Additional comments

no comment

·

Basic reporting

The authors didn’t completely address my concerns. Species is misused throughout the whole manuscript. Species is a taxonomic level, as well as genus or phylum.

Figure numbers are messed up. Please correct throughout the manuscript.

Line 211-218: Please provide references for each primer used to amplify 16S rRNA gene and fungal ITS. Change ‘priors’ or ‘prims’ to ‘primers’.

Lines 223-227: Please provide what package or software and what version were used to process raw sequences for generating OTU table. Delete ‘Amplicon Sequence Variants’. ASVs were obtained with processing method differed from OTU. What database and what version were used to annotate sequences.

Lines 230: Change ‘species diversity’ into ‘microbial diversity’. SPECIES IS A TAXONOMY LEVEL. PLEASE TAKE CAUTION WITH USING ‘SPECIES’ THROUGHOUT THE MANUSCRIPT. NO SPECIES HAS BEEN DETECTED IN THE STUDY.

Lines 232-233: PCA analysis was used to determine the distribution pattern of samples, based on the microbial community composition. Microbial diversity is usually about alpha diversity. Beta diversity is about the difference between two samples/two microbial community compositions/two ecosystems. Please rephrase this sentence.

Lines 305-311: Please explain what are ‘2459 features’ or any features here. Please double check if ‘features’ are ‘OTUs’. If so, correct it THROUGHOUT THE MANUSCRIPT.

Lines 360-363: Fig 10 didn’t show that F1+B could enhance the abundance of any dominant fungal phyla, compared to CK and F1. If so, please indicate what dominant fungal phyla was enriched in F1+B. Change ‘genus level’ to ‘phylum level’.

Line 367-369: Change ‘under F1+B treatment’ to ‘under F1+B and F1 treatments’. Please indicate what specific genera here for Basidiomycota (decreased) or Mortierellomycota (increased) in Fig 11.

Lines 375-376: Delete ’and promotes the evolution of the soil fungal community’.

Experimental design

No comments

Validity of the findings

No comments

---

## Round 0.3 · Minor Revisions

Dear Authors,
Please inform about the reference and database as mentioned by the reviewer.

·

Basic reporting

Lines 213-218: Reference [30] is a paper about sequence processing algorithm DADA2. Please provide references for each primer used in the study.

Line 226: Please provide what database (e.g. Silva, Greengenes, etc.) and what version were used to annotate sequences.

Experimental design

no comment

Validity of the findings

no comment

---

## Round 0.4 · accepted · Accept

The authors have addressed all of the reviewers' comments.This manuscript is ready for publication.

·

Basic reporting

No comments.

Experimental design

No comments.

Validity of the findings

No comments.